# Switching to a Standard Chow Diet at Weaning Improves the Effects of Maternal and Postnatal High-Fat and High-Sucrose Diet on Cardiometabolic Health in Adult Male Mouse Offspring

**DOI:** 10.3390/metabo12060563

**Published:** 2022-06-18

**Authors:** Andrea Chiñas Merlin, Kassandra Gonzalez, Sarah Mockler, Yessenia Perez, U-Ter Aondo Jia, Adam J. Chicco, Sarah L. Ullevig, Eunhee Chung

**Affiliations:** 1Department of Kinesiology, University of Texas at San Antonio, San Antonio, TX 78249, USA; andreacmerlin@gmail.com (A.C.M.); kassglz127@gmail.com (K.G.); mockler@livemail.uthscsa.edu (S.M.); yessenia.perez@utsa.edu (Y.P.); u-teraondo.jia@utsa.edu (U.-T.A.J.); 2Biomedical Engineering, Tecnologico de Monterrey, Campus Monterrey, Monterrey 64849, Mexico; 3Department of Biomedical Sciences, Colorado State University, Fort Collins, CO 80523, USA; adam.chicco@colostate.edu; 4College for Health, Community and Policy, University of Texas at San Antonio, San Antonio, TX 78249, USA; sarah.ullevig@utsa.edu

**Keywords:** diabetes, obesity, male mouse offspring, glucose and insulin tolerance tests, systolic function

## Abstract

Cardiac mitochondrial dysfunction contributes to obesity-associated heart disease. Maternal and postnatal diet plays an important role in cardiac function, yet the impacts of a mismatch between prenatal and postweaning diet on cardiometabolic function are not well understood. We tested the hypothesis that switching to a standard chow diet after weaning would attenuate systemic metabolic disorders and cardiac and mitochondrial dysfunction associated with maternal and postnatal high-fat/high-sucrose (HFHS) diet in mice. Six-month-old male CD1 offspring from dams fed a HFHS diet and weaned to the same HFHS diet (HH) or switched to a standard chow diet (HC) were compared to offspring from dams fed a low-fat/low-sucrose diet and maintained on the same diet (LL). HC did not decrease body weight (BW) but normalized glucose tolerance, plasma cholesterol, LDL, and insulin levels compared to the HH. Systolic function indicated by the percent fractional shortening was not altered by diet. In freshly isolated cardiac mitochondria, maximal oxidative phosphorylation-linked respiratory capacity and coupling efficiency were significantly higher in the HC in the presence of fatty acid substrate compared to LL and HH, with modification of genes associated with metabolism and mitochondrial function. Switching to a standard chow diet at weaning can attenuate the deleterious effects of long-term HFHS in adult male mouse offspring.

## 1. Introduction

Obesity is an independent risk factor for heart disease. The prevalence of childhood obesity has increased at a startling rate over the past decades in parallel with the overweight and obesity rate in childbearing women [1]. Systemic reviews of animal studies demonstrate that pre-pregnancy, gestational obesity, and maternal high-fat diet are strong predictors of childhood obesity, metabolic complications, and a higher risk of cardiovascular diseases in offspring [1,2]. Further, continuous high-fat diet consumption through gestation, lactation, and postweaning worsens lipid biomarkers, glucose intolerance, and cardiac contractile function compared to the offspring exposed to a high-fat diet after weaning, but not during the gestation and lactation period [3]. In contrast, several studies raise interesting notions that the mismatch between prenatal and postnatal diet may elicit different health outcomes in rodent offspring [4]. For example, some studies demonstrate that a maternal high-fat diet elicits protective effects for offspring when the offspring are challenged with a high-fat diet when they reach young adults such as improved glucose tolerance, insulin sensitivity, fasting cholesterol, and endothelial function in rat male offspring [5,6]. In addition, switching to a low-fat diet from a prenatal high fat diet provides better metabolic outcomes than in mouse offspring fed a low-fat diet continuously [4].

Mitochondrial dysfunction is highly linked to metabolic disorders and cardiac dysfunction associated with obesity and diabetes [7,8]. Genetically obese mice (ob/ob) showed decreased mitochondrial oxidative capacity and mitochondrial efficiency with fatty acid substrates but not glucose substrates [7]. Rodent offspring born from maternal diabetes [8] and obesity [9] have significantly decreased cardiac mitochondrial function. However, the mismatch between prenatal and postnatal diet challenges in offspring on cardiac mitochondrial function has not been studied. In addition, maternal obesity or maternal nutritional stress with a Western-style diet without obesity has elicited contradictory results in offspring health outcomes [2]. Thus, the objective of this study was to investigate the effects of maternal and postnatal diet on glucose metabolism, cardiac function, and mitochondrial respiration in male mouse adult offspring born from maternal HFHS dams without obesity, which allowed us to examine the diet effect alone [10]. We tested the hypothesis that switching to a standard chow diet at weaning would attenuate systemic metabolic disorders and cardiac and mitochondrial dysfunction associated with maternal and postnatal HFHS diet in adult male mouse offspring.

## 2. Results

### 2.1. Switching to a Standard Chow Diet at Weaning from Maternal HFHS did Not Alter Physical Characteristics

To study maternal diet effects alone, independent of obesity, the male offspring from obesity-resistant dams fed HFHS were used. Body weight (BW) was significantly higher in offspring born from HFHS compared to those from LFLS at weaning (Figure 1A) and continued to be higher during their growth phase (Figure 1B). The weight gain was similar in offspring from dams fed a HFHS diet whether they were weaned to the same HFHS diet (HH) or switched to a standard chow diet at weaning (HC). The weight gain was significantly different in offspring from dams fed a low-fat/low-sucrose diet and maintained on the same diet (LL) as compared to HH as well as LL compared to HC (Figure 1B). The final BW was significantly lower in LL than HH or HC but was similar between HH and HC (Figure 1C). However, while fat mass was higher in HH compared to LL, it was not significantly different between LL and HC (Figure 1D). LL mice had a shorter tibial length (TL) than HH or HC (Figure 1E). Our results demonstrate that weaning offspring to a standard diet from maternal HFHS did not normalize BW and growth trajectory.

### 2.2. Postweaning Control Diet Improved Glucose and Insulin Tolerance Tests

Blood glucose levels at postnatal day 3 showed no difference between offspring born from LFLS or HFHS (100.4 ± 7.0 mg/dL, *n*= 11 in LFLS vs. 116.4 ± 6.2 mg/dL, *n* = 12 in HFHS). GTT showed that a persistent HFHS diet after weaning (HH) impaired glucose clearance when compared to LL, while switching to the control diet (HC) improved it (Figure 2A). The GTT area under the curve (AUC) was significantly higher in HH compared to LL, but no differences were found between LL and HC (15,434 ± 2461 in LL; 24,672 ± 2321 in HH; 18,751 ±1434 in HC) (Figure 2B). In terms of ITT, the mice in the HH group developed insulin resistance indicated by slowed glucose clearance after insulin administration and larger ITT AUC compared to LL (Figure 2C,D). In contrast, there were no differences in glucose clearance between LL and HC, indicating preservation of insulin sensitivity with a postweaning standard diet (HC) (Figure 2D). However, there was no difference between HH and HC in ITT AUC due to large animal variations (12,961 ± 1204 in LL; 21,633 ± 3790 in HH; 16,384 ± 2690 in HC). Overall, a postweaning diet resulted in improved glycemic control.

### 2.3. Postweaning Control Diet Improved Hyperlipidemia and Hyperinsulinemia

A post-weaning HFHS diet coupled with exposure to the maternal HFHS diet (HH) changed several lipid biomarkers and hormones (Table 1). Total cholesterol (TC) and low-density lipoprotein cholesterol (LDL) were significantly higher in the HH compared to the LL, but significantly lower in the HC compared to the HH. There were no differences between the LL and HC groups. The levels of non-esterified free fatty acid (NEFA) and triglyceride (TG) were similar among the groups. Insulin and leptin levels were significantly higher in HH compared to LL. Insulin levels were returned to the levels of LL in the HC mice, but leptin levels were significantly higher in HC compared to LL resulting in no differences between HH and HC. Offspring weaned to HC deviated from maternal HFHS improved hyperlipidemia (TC and LDL) and hyperinsulinemia, but not leptin levels.

### 2.4. Cardiac Systolic Function and Molecular Markers of Pathological Cardiac Hypertrophy Were Not Altered by Either Maternal or Postnatal Diet

The absolute heart weight (HW) was significantly higher in HH than in LL but was not different between LL and HC or HH and HC (Figure 3A). The relative heart weight, HW normalized to BW, was significantly higher in LL compared to HH or HC due to heavier BW (Figure 3B). HW normalized to TL (HW/TL), commonly used for the cardiac hypertrophic index, was significantly higher in HH than in LL (Figure 3C).

Echocardiographic assessment of the left ventricle (LV) revealed that the LV structure and systolic function were similar among groups, except for LV diameter (Table 2). The LV diameter at diastole (LVIDd) and systole (LVIDs) was significantly higher in HH compared to LL. Relative wall thickness (RWT), often used for cardiac hypertrophy index from the echocardiographic assessment [11], was similar among groups. Although the cardiac dimension was elevated in the HH group, the percent fractional shortening (%FS) and the percent ejection fraction (%EF) were not altered by diet (Table 1). Next, we measured genes typically upregulated in pathological cardiac hypertrophy: genes regulating collagen synthesis (*Col1a1*, *Col3a1*, *Col8a1*, and *Ccl2*) were not altered by diet (Figure 3D). In addition, genes dominantly expressed in the fetal heart (*Myh6*, *Myh7*, *Pln*, *Atp2a2*, *Acta1*, *Nppa*, and *Nppb*) but re-expressed during pathological stimuli were also not altered (Figure 3E). We measured protein levels of the myosin heavy chain since increased beta-myosin heavy chain (β-MyHC) leads to contractile dysfunction [12]. The ratio between β-MyHC/α-MyHC was not altered among groups (Figure 3F,G). Our results demonstrate that diet had no effect on systolic function and did not induce pathological gene expression in the hearts of adult male mouse offspring.

### 2.5. Postweaning Control Diet Modulates the Cardiac Mitochondrial Respiratory Function

Electron leak state in the presence of malate and pyruvate was not different among groups (Figure 4A). Complex I respiration measured by adding ADP (state 3 respiration) was also not different among groups (Figure 4B). However, complex I + II-supported oxidative phosphorylation (OXPHOS) by further addition of succinate was significantly higher in HC compared to HH (Figure 4C). The coupling efficiency with pyruvate (Figure 4D) was not different among groups. However, in the presence of fatty acid substrate (Malate + Palmitoyl-L-carnitine), the LEAK respiration (Figure 4E), ADP-stimulated respiration rates supported by pyruvate and palmitoylcarnitine (FAOp, Figure 4F), the CI-dependent respiration rate (PMP-FAO, Figure 4G), and maximal OXPHOS-linked respiration rate (CI + CIIp FAO, Figure 4H) were all significantly higher in HC compared with HH. OXPHOS coupling control factors in the presence of fatty acids (FAO(P-L)/P, Figure 4I) were also significantly higher in HC compared to HH. Interestingly, no differences were found between LL and HH in all parameters of mitochondrial respiratory function. Citrate synthase (CS) activity, a common biomarker for mitochondrial density [13] (Figure 4J), and cytochrome c oxidase activity (Figure 4K), a marker of mitochondrial cristae surface density [14], were similar among groups. Thus, the improved respiratory function with fatty acid substrates shown in HC was not due to increased mitochondrial density or mitochondrial cristae surface area.

### 2.6. Postweaning Control Diet Alters Genes Regulating Mitochondrial Function and Metabolism

Next, we determined whether altered mitochondrial efficiency is linked to the genes regulating mitochondrial function and cardiac metabolism. *Nrf1* and *Nrf2*, which regulate mitochondrial biogenesis, were significantly downregulated in the HC compared to the LL. *Mfn1*, a gene that regulates mitochondrial dynamics, was significantly downregulated in the HC compared to LL. *Nrf1*, *Mfn1*, and *Opa1* were significantly downregulated in the HC compared to HH (Figure 5A). Among genes regulating cardiac metabolism, *Cpt1b* and *Glut 4* were significantly downregulated in HC compared to LL. *Cpt2* and *Cd36* were significantly downregulated in HC compared to HH (Figure 5B). Interestingly, there were no alterations between LL and HH.

### 2.7. Proteins Regulating Fatty Acid Metabolism Are Altered with Maternal and Postweaning Diet

Stearoyl-Coenzyme A desaturase 1 (SCD1), a key protein for regulating fat deposition and composition, was significantly increased in HH compared to LL but was not different between HH and HC (Figure 6A). UCP3, a major isoform of uncoupling protein expressed in cardiac muscle that is known to regulate metabolism and ROS production, was significantly increased in the HH compared to LL and HC (Figure 6B). SOD2, a major mitochondrial antioxidant enzyme, was significantly increased in HC compared to LL (Figure 6C). Phosphorylation of AMPK relative to AMPK, the important signaling pathways regulating energy homeostasis, was significantly decreased in the HC compared to the LL as well as HH compared to LL (Figure 6C). The MDA level, a marker of lipid peroxidation, was significantly higher in HC compared to HH or LL.

## 3. Discussion

This study is the first to comprehensively evaluate the effects of maternal and postnatal diet on physical characteristics, lipid biomarkers, GTT, ITT, systolic function, cardiac and mitochondrial respiratory function, and molecular markers of cardiac function in adult male mouse offspring. Our results demonstrate that the detrimental effects of a persistent energy-dense nutritional environment through gestation, lactation, and postweaning can be improved by switching to a standard chow diet after weaning in male offspring. The observed positive effects include normalized glucose tolerance, plasma TC, LDL, and insulin levels to the LL group, improved efficiency of mitochondrial fatty acid oxidation, and downregulation of genes regulating fatty acid metabolism (*Cpt1b*, *Cpt2*, and *Cd36*), mitochondrial biogenesis (*Nrf1*), and dynamics (*Mfn1* and *Opa1*).

Postweaning HC diet did not alleviate BW and fat mass associated with maternal and postnatal HFHS, which supports previous studies demonstrating the importance of the intrauterine environment on the metabolic health of offspring, such that offspring born from mothers fed an obesogenic diet have a higher chance of becoming obese even when offspring were fed a low-fat diet or standard chow diet at weaning [3,15]. Switching to the standard chow diet at weaning from HFHS (HC) had less effect on BW and fat mass although HC normalized plasma TC, LDL, and insulin levels to the LL levels. The BW and adiposity may be largely influenced by circulating leptin levels [16] or decreased leptin sensitivity [15] which can be programmed during the developmental phase. Our results reinforce the importance of the intrauterine environment in regulating BW and adiposity in male offspring.

Cardiac function has been shown to be affected by diet. An energy-dense diet feeding either HF lard diet or HFHS significantly decreased systolic function in male mice [17]. Eight-week-old male offspring born from maternal obesity significantly decreased %FS independent of postweaning diet, whether weaned to HC or maternal obesogenic diet [18]. Previously, 12-week-old male offspring born from obese dams but weaned onto a standard diet showed significantly decreased systolic (i.e., rate of pressure generation and LV developed pressure) and diastolic (i.e., increased LV end-diastolic pressure) parameters measured by isolated Langendoff heart perfusion techniques [19]. In contrast, we found no changes in systolic function indicated by %FS or %EF in our studied animals. The contradicting results could be due to differences in diet composition as our diet was composed of 45% fat and 34% sugar, while Loche et al. [18] included sweetened condensed milk (55% sugar, 8% fat kcal/g) in addition to the 10% sugar and 20% fat from the assigned obesogenic diet which could have led to more significant changes in %FS. Second, the age and the strain of the mice were different: our mice were CD1 and studied at a later age (25 weeks old), while previous studies were younger C57BL/6 mice: 8 weeks [18] and 12 weeks [19]. Third, our mice in HH and HC were born from dams fed HFHS but not obese [10], in contrast to offspring in the previous studies which were from obese dams [18,19]. No induction of fetal genes and no changes in contractile proteins, such as myosin heavy chain composition, support our functional data. However, we cannot rule out that diastolic function may change in our studied animals since diabetic cardiomyopathy leads to diastolic dysfunction with preserved systolic function [20]. Thus, future studies are warranted to investigate diastolic function because mice in the HH group displayed significantly impaired GTT and ITT compared to the LL group.

Cardiac lipotoxicity through altered fatty acid and glucose metabolism could lead to mitochondrial dysfunction. In contrast to our hypothesis, mitochondrial function was not altered in HH compared to LL, nor were genes regulating mitochondrial function and metabolism. We do not have a clear explanation, but we may speculate on several aspects: (1) A population-based prospective cohort study demonstrates that gestational weight gain is a strong predictor of the cardiometabolic health of offspring [21] and the absence of gestational weight gain in female mice could account for the minimal effects seen in the immediately subsequent generation [22]. Since dams of LL, HH, and HC were similar in BW, gestational weight gain, glucose tolerance, and lipid biomarkers [10], differences between LL and HH may not be observed. (2) Increased UPC3 protein levels in the hearts of HH may provide protective effects [23] that reduce reactive oxygen species production [24] which was shown here with no increase in lipid peroxidation. (3) Although it is generally acknowledged that contradicting results of offspring outcomes in response to maternal nutritional stress are largely due to the different composition of fat content, less attention was given to the control diets [25]. The importance of a purified control diet that matches minerals, vitamins, and the source of nutrients was emphasized when the gut microbiome was investigated as the mechanism of obesity and other diseases [25]. However, lower fat content (i.e., 10%) may not be sufficient to support healthy pregnancies [26] although many diet-induced obesity studies use a refined low-fat diet with 10% kcal from fat [27]. Considering the importance of fat content to maintaining healthy pregnancy [28], the dietary fat content was increased in this study similar to AIN-93 (~17% kcal from fat) [26], which was the level recommended for pregnancy. However, a long duration of feeding refined LFLS (i.e., from gestation, lactation, and postweaning) without a portion of dietary fermentable fibers, which can be found in a standard chow diet, could modify adiposity [29] and may elicit negative health outcomes as shown previously in fatty livers induced by LFD in female mice [30]. Despite studies conducted by Almeida-Suhett et al. [31] pointing out that regular chow and purified LFD have similar effects on metabolic and behavioral outcomes in male mice, LFD during pregnancy may act as nutritional stress in both dam and offspring. Future studies warrant comparing regular chow and refined LFD on pregnancy outcomes and the metabolic health of offspring.

Emerging recent studies demonstrate that it is not the maternal environment per se, but rather the mismatch between the prenatal and postnatal environment that predicts the health and disease risk which is termed the predictive adaptive response (PAR) hypothesis coined by Gluckman and Hanson [32]. Exposure to fat feeding during pregnancy can provide metabolic advantages in the offspring [4,5,6], which may explain the results of HC mice demonstrating improved mitochondrial respiratory function and efficiency in the presence of fatty acid substrates since the heart relies heavily upon efficient fatty acid oxidation to meet its energy demands. The loss of this efficiency has been associated with diabetic cardiomyopathy [33]. Overexpression of SCD1 inhibits fatty acid oxidation and apoptosis through an inhibition of AMPK, which protects the heart from lipotoxicity [34], which we showed in the heart of HC. In addition to increased SCD1 levels, downregulation of key genes regulating fatty acid metabolism such as *Cpt1b,* the rate-limiting enzyme in long-chain fatty acid oxidation [35]; *Cpt2*, an essential enzyme for fatty acid oxidation within mitochondria [36]; and *Cd36*, a fatty acid transport protein [37], may be beneficial in protecting the heart from lipotoxicity. Intriguingly, MDA levels (a marker for lipid peroxidation) were significantly higher in HC compared to HH and LL, although a previous study also showed decreased MDA levels in the hearts of non-obesogenic HFD-fed mice compared to the control diet [38]. SOD2, mitochondrial antioxidant enzyme levels were significantly higher in HC compared to LL, but UCP3, previously demonstrated as an antioxidant defense system by reducing ROS production [39], was significantly lower in HC compared to HH. The deficiency of *Nrf1* and *Nrf2* induces oxidative stress [40], while *Opa1* protects cells from apoptosis [41]. Significantly decreased *Nrf2* and *Opa1* in HC compared to HH may alter the antioxidant defense system and ROS production. Switching diets to reduce fat availability may reduce fatty acid transport and also may affect mitochondrial morphology which might increase lipid peroxidation in the hearts of HC. A future study is warranted to measure ROS production simultaneously with mitochondrial respiration.

There are some limitations to consider. First, we did not include a postweaning control diet in offspring born from LFLS, which limited determining the maternal diet effect independent of the postweaning diet. Second, we did not switch diet to LFLS; instead, we used a standard chow diet immediately postweaning in offspring born from HFHS dams.

In summary, our results provide important insights into the cardiometabolic health of offspring from pregnancies associated with maternal overnutrition. Feeding offspring a healthy diet postweaning following a maternal HFHS may provide metabolic advantages that decrease the risk of cardiometabolic disease later in life. Improvements in circulating LDL, glucose tolerance and insulin sensitivity, and insulin are also likely to confer significant reductions in cardiometabolic risk, along with reduced fat mass despite elevated BW. Finally, future studies warrant measuring reactive oxygen species (ROS) production and antioxidant enzyme defense mechanisms to explain increased lipid peroxidation in the HC with increased mitochondrial efficiency.

## 4. Materials and Methods

### 4.1. Experimental Group

Detailed physiological characteristics of dams, breeding schemes, and diet information were published previously [10]. Briefly, nulliparous female CD-1 mice were fed either a high-fat, high-sucrose diet (HFHS, 4.7 kcal/g with 45%, 15%, and 40% total calories from fat, protein, and carbohydrate, respectively, with 34% sucrose wt:wt, TD.08811, Envigo, Madison, WI, USA) or a refined low-fat, low-sucrose diet (LFLS, 3.8 kcal/g with 17%, 18%, and 64% total calories from fat, protein, and carbohydrate, respectively, with 12% sucrose wt:wt, TD.170522, Envigo, Madison, WI, USA) for eight weeks before initiation of pregnancy, during gestation, and lactation. Following 8 weeks of HFHS diet, half of the mice were obese and the other half were obese resistant. In this study, we used male offspring born from obese resistant dams [10] to determine maternal diet effect independent of maternal obesity. At postnatal day 21, all male mice were weaned into 3 groups: offspring born from maternal LFLS and postnatal LFLS (LL), maternal HFHS and postnatal HFHS (HH), and maternal HFHS weaned onto a standard laboratory chow diet (HC; PicoLab Select Rodent 50 IF/6F, 5V5R, Lab Diet, Fort Worth, TX, USA), all studied at 5–6 months of age. One male mouse from each litter was used for this study. Glucose tolerance test (GTT) and cardiac function were determined at 5 months of age. All experiments were carried out following the relevant guidelines and regulations (protocol # MU109) approved by the University of Texas at San Antonio Institutional Animal Care and Use Committee. The study was reported under ARRIVE guidelines [42]. The experimental scheme was shown in Figure 7.

### 4.2. Intraperitoneal Glucose—And Insulin Tolerance Tests

5-month-old mice were subjected to a 5 h fast while still granted free access to water for glucose tolerance test (GTT) and insulin tolerance test (ITT). After the fasting period, the mice were weighed and basal blood glucose level (0 min) was measured using a glucometer (AimStrip Plus, VWR, Radnor, PA, USA). For GTT, following the baseline measurement, 2 g/kg body weight of glucose (20% D-glucose dissolved in PBS) was administered by intraperitoneal injection. Blood glucose levels were measured from the tail vein at 15, 30, 60, and 120 min post-injection. The trapezoidal method was used to calculate the total area under the curve (AUC). ITT was performed similarly to the GTT with an intraperitoneal injection of 1 U/kg body weight of insulin after baseline glucose measurement (Humulin R, 100 U/mL, Henry Schein Animal Health, Dublin, OH, USA). Blood glucose levels were measured at 15, 30, 60, and 120 min after injection. The trapezoidal method was used to calculate the total area under the curve (AUC). To reduce stress on the animals, there was a one-week wash-out period between GTT and ITT.

### 4.3. Assessment of Cardiac Function

Systolic function and ventricular dimensions were measured using echocardiography on 5-month-old offspring. Mice were anesthetized with inhaled isoflurane (1.5% maintenance after 3% induction, SomnoSuite, Kent Scientific, Torrington, CT, USA) and measured at the midpapillary short-axis views using M-mode echocardiography equipped with a 10–22 MHz transducer (NEXTGen LOGIQ e R7, GE, Wauwatosa, WI, USA). All measurements were obtained three consecutive cardiac cycles. Relative wall thickness [(IVSd + LVPWd)/LVIDd] was calculated as previously described [11].

### 4.4. Sample Collection

Male offspring were euthanized for tissue collection at 6 months of age. The blood samples were collected by cardiac puncture from anesthetized mice with isoflurane by inhalation and prepared the plasma samples as described previously [10]. Hearts were rapidly excised and dissected out of the left ventricle. Half of the left ventricle was used right away for mitochondrial isolation by differential centrifugation as described previously. The other half of the left ventricle was immediately frozen in liquid nitrogen and stored at −80 °C for further analysis. Both inguinal and epididymal fat were measured to quantify fat mass.

### 4.5. Plasma Analyses

Colorimetric assays were used to evaluate lipid biomarkers levels, including low-density lipoprotein cholesterol (LDL), non-esterified fatty acids (NEFA), total cholesterol (TC), and triglycerides (TG) (Wako Chemical, Richmond, VA, USA). Plasma insulin levels were measured using mouse enzyme-linked immunosorbent assays kits (EZRMI-13K, EMD Millipore Co., Billerica, MA, USA). All assays were performed according to the manufacturer’s instructions.

### 4.6. Assessment of Mitochondrial Respiratory Function

Mitochondrial respiratory function was assessed from freshly isolated mitochondria from the left ventricle by high-resolution respirometry (Oxygraph-2k, Oroboros Instruments, Innsbruck, Austria) as described previously with some modifications [43]. Briefly, 30 μg of mitochondrial protein from the left ventricle were added to the respiration chamber containing 2 mL of mitochondrial respiration medium (MiR05: 0.5 mM EGTA, 3 mM MgCl2 hexahydrate, 60 mM lactobionic acid, 20 mM taurin, 10 mM KH2PO4, 20 mM HEPES, 110 mM Sucrose, and 0.1% BSA, pH 7.1 with KOH). Two distinct mitochondrial substrate titration protocols were used to measure complex I and II respiration mediated by fatty acid with carbohydrate protocol and carbohydrate substrates only. For carbohydrate protocol, the following were added sequentially: 1 mM malate + 5 mM pyruvate (M + Pyr), 2.5 mM ADP, 10 mM glutamate, and 20 mM succinate. For fatty acid with carbohydrate protocol, 1 mM malate + 0.025 mM palmitoyl-L-carnitine (M + Pal), 2.5 mM ADP, 5 mM pyruvate, 10 mM glutamate, and 20 mM succinate were added. These protocols determined respiratory LEAK (L), oxidative phosphorylation (OXPHOS) capacities (P) in the presence of malate and pyruvate (M + P) and fatty acid (M + Pal), OXPHOS coupling control (MPP-L/P). In both protocols, 10uM of cytochrome c followed by succinate was added to test membrane integrity. Data were collected with Datlab 7 software (Oroboros, Innsbruck, Austria). Respiration rates (nmol mg^−1^ sec^−1^) were normalized per mg of mitochondrial protein determined by BCA assay (Pierce BCA Protein Assay Kit, ThermoFisher Scientific, Waltham, MA, USA).

### 4.7. Biochemical Assays

The activities of two major mitochondrial enzymes, citrate synthase (CS0720; Sigma-Aldrich, St. Louis, MO, USA) and cytochrome c oxidase (CYTOCOX1; Sigma-Aldrich, St. Louis, MO, USA) were measured using isolated mitochondria from the left ventricle that were used for mitochondrial respiratory function. Lipid peroxidation was determined by measuring malondialdehyde (MDA) levels in Thiobarbituric Acid Reactive Substances (TBARS) assay (Cayman Chemical, Ann Arbor, MI, USA). All assays were carried out according to the manufacturer’s instructions.

### 4.8. Quantitative Real-Time PCR (qRT-PCR)

Cardiac gene expression was determined according to the protocol described previously [10]. The following genes were determined: (1) gene-associated pathological cardiac hypertrophy (α-myosin heavy chain (*Myh6*), beta-myosin heavy chain (*Myh7*), phospholamban (*Pln*), sarcoplasmic reticulum Ca2+ ATPase 2a (*Atp2a2*), α-skeletal actin (*Acta1*), atrial natriuretic peptide (*Nppa*), and brain natriuretic peptide (*Nppb*)); (2) interstitial collagen content collagen type 1 alpha chain (*Col1a1*, *Col3a1*, and *Col8a1*); (3) *Ccl2*, also known as MCP-1, an important proinflammatory chemokine; (4) glucose transport (*Glut4* and *Irs-1*); (5) gluconeogenesis (*Pepck*), fatty acid transport and oxidation (*Cd36*, *Cpt* 1, *Cpt2*, and *Mcad*); (6) genes regulating mitochondrial biogenesis (*Ppargc1a*, *Tfam*, *Nrf1*, *Nrf2*); and (7) genes regulating mitochondrial dynamics (*Drp1*, *Mfn1*, *Mfn2*, *Opa1*, *Cypd*). The common reference genes, such as *Gapdh*, *Actb*, *36B4*, *B2M*, *Tbp*, and *Hprt*, were evaluated using M-value, the reference gene expression stability using Bio-Rad CFX Manager 3.2. (Biorad, Hercules, CA, USA). *B2M* was most stable among groups, so mRNA levels were normalized to *B2M* using the cycle threshold (ΔΔCT), and relative fold changes were reported compared to the LL. Primers used for this study are listed in Appendix A.

### 4.9. Protein Analyses

Left ventricular tissues were homogenized using the radio-immunoprecipitation assay (RIPA) buffer (25 mM Tris∙HCl pH 9.4, 150 mM NaCl, 1% NP-40, 1 mM EDTA, and 5% glycerol) (Sigma-Aldrich, St. Louis, MO, USA) containing protease and phosphatase inhibitor single-use cocktail (ThermoFisher Scientific, Waltham, MA, USA). Following homogenization, samples were centrifuged at 10,000× *g* for 10 min at 4 °C, and the supernatant was transferred to a tube, and stored at −80 °C. Western blot was conducted according to the protocol described previously, in detail [44]. Briefly, 30 μgs of tissue homogenates was used, and the following proteins were evaluated: SCD1 (2794, Cell Signaling, Danvers, MA, USA), SOD2 (13141, Cell Signaling, Danvers, MA, USA), Anti-Vinculin (PA5-29688, ThermoFisher Scientific, Waltham, MA, USA), UCP3 (#PA1-055, Invitrogen, Waltham, MA, USA), AMPKα (2532, Cell signaling, Danvers, MA, USA), p-AMPKα (2535, Cell signaling, Danvers, MA, USA). Vinculin was used for loading control.

The content of myosin heavy chain isoform (MyHC) of ventricular homogenates was determined as previously described with modification [10]. Briefly, 15 μg of homogenates were loaded into Acylamide- N, N’-Diallyl-L-tartardiamide (DATD) gels and ran for 4 h at 16 mA at 4 °C. Four to five animals per group were used with three technical replicates per animal, and a representative gel image was shown.

### 4.10. Statistical Analyses

Data are presented as group mean ± standard error of the mean (SEM) with the number of samples per group given in the figure legends and tables. Group differences were compared by one-way analysis of variance (ANOVA) or two-way ANOVA with post hoc Fisher’s LSD tests. The statistical significance was set at *p* < 0.05 for all analyses. All statistical analyses were conducted using GraphPad Prism software version 9.0 (GraphPad, San Diego, CA, USA).

## 5. Conclusions

A negative intrauterine environment is a strong predictor of obesity and metabolic disorders in offspring. Mitochondrial dysfunction is highly linked to metabolic disorders. Feeding offspring a standard chow diet at postweaning can provide a metabolic advantage by improving mitochondrial fatty acid oxidation, glucose tolerance, and cholesterol and insulin levels following the maternal high-fat, high-sucrose diet. Further investigation to determine if healthy diets in infants and children can counteract a negative intrauterine environment is warranted.

## Figures and Tables

**Figure 1 metabolites-12-00563-f001:**
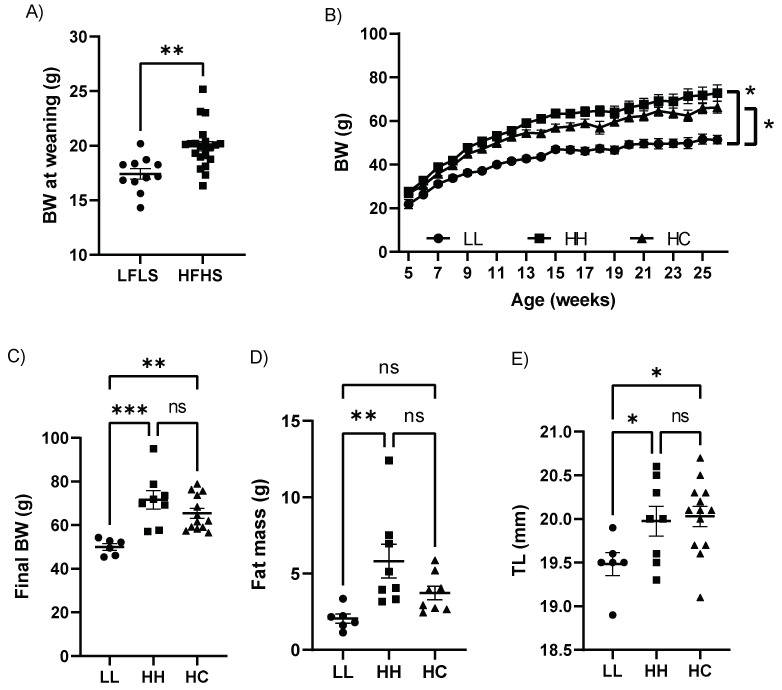
Physical characteristics. (**A**) BW at weaning; (**B**) BW trend over time; (**C**) Final BW; (**D**) Fat mass; and (**E**) TL in LL, HH, and HC mice. BW, Body weight; TL, Tibial length; HFHS, a high-fat and a high-sucrose diet; LFLS, a low-fat and a low-sucrose diet; LL, offspring born from maternal LFLS and postnatal LFLS indicated by closed circles (●); HH, maternal HFHS and postnatal HFHS indicated by closed squares (■); HC, maternal HFHS weaned onto a standard laboratory chow diet indicated by a closed triangle (▲). Values are expressed as mean ± SEM. * *p* < 0.05; ** *p* < 0.01; *** *p* < 0.001; ns, not significant. *n* = 6 in LL; *n* = 8 in HH; and *n* = 8–13 in HC.

**Figure 2 metabolites-12-00563-f002:**
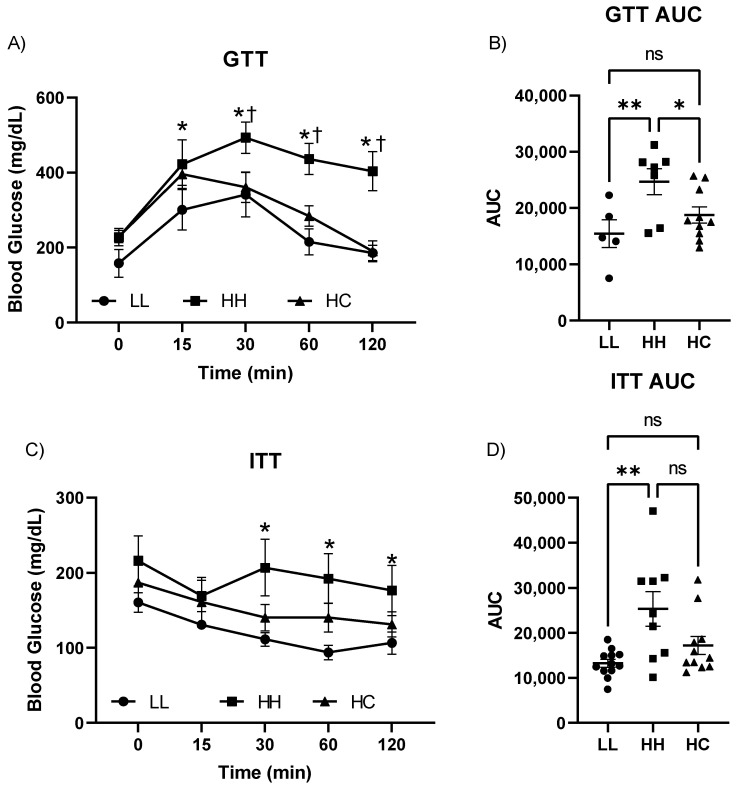
Intraperitoneal glucose and insulin tolerance test. (**A**) GTT; (**B**) GTT AUC; (**C**) ITT; and (**D**) ITT AUC in LL, HH, and HC mice. GTT, glucose tolerance test; AUC, area under the curve; ITT, insulin tolerance test; LL, offspring born from maternal LFLS and postnatal LFLS indicated by closed circles (●); HH, maternal HFHS and postnatal HFHS indicated by closed squares (■); HC, maternal HFHS weaned onto a standard laboratory chow diet indicated by a closed triangle (▲). Values are presented as mean ± SEM. Statistical significance is calculated by one-way ANOVA (**B**,**D**) or two-way ANOVA with post hoc Fisher’s LSD comparisons (**A**,**C**). * *p* < 0.05 and ** *p* < 0.01 vs. LL; † *p* < 0.05 vs. HH; ns, not significant. *n* = 6–12 in LL; *n* = 8–9 in HH; *n* = 11 in HC.

**Figure 3 metabolites-12-00563-f003:**
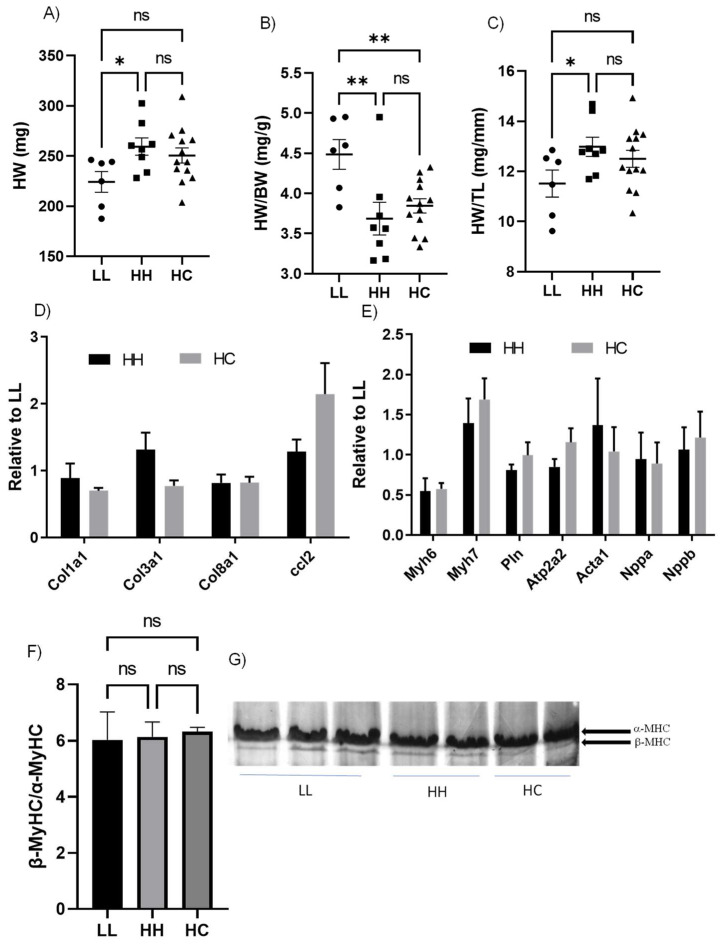
Cardiac adaptation associated with diets. (**A**) Absolute HW; (**B**) Relative HW normalized to BW; (**C**) HW normalized by TL; (**A**–**C**) *n* = 6 in LL; *n* = 8 in HH; and *n* = 8–13 in HC; (**D**) genes regulating extracellular matrix; (**E**) Genes associated with pathological cardiac hypertrophy; (**D**,**E**) *n* = 4–8 samples/group and fold changes relative to LL are shown. mRNA levels were normalized to B2M; (**F**) Bar graph represents the β-MyHC to α-MyHC ratio. *n* = 4–5/group with five technical replicates per animal; and (**G**) Representative gels showing myosin heavy chain (MyHC) isoforms in LV homogenates from LL, HH, and HC; HW, heart weight; BW, body weight; TL, tibial length; *Col1a1*, collagen type 1 alpha chain; *Col3a1*, collagen type III alpha 1 chain; *Col8a1*, collagen type VIII alpha 1 chain; *Ccl2*, C-C motif chemokine ligand 2; *MyH*, myosin heavy chain; *Pln*, phospholamban; *Atp2a2*, cardiac sarcoplasmic reticulum Ca2+ ATPase 2a; *Acta1*, alpha-skeletal muscle actin; *Nppa*, atrial natriuretic peptide; *Nppb*, brain natriuretic peptide; LL, offspring born from maternal LFLS and postnatal LFLS indicated by closed circles (●); HH, maternal HFHS and postnatal HFHS indicated by closed squares (■); HC, maternal HFHS weaned onto a standard laboratory chow diet indicated by a closed triangle (▲). Values are presented as mean ± SEM. Statistical significance is calculated by one-way ANOVA with post hoc Fisher’s LSD comparisons. * *p* < 0.05; ** *p* < 0.01; ns, not significant.

**Figure 4 metabolites-12-00563-f004:**
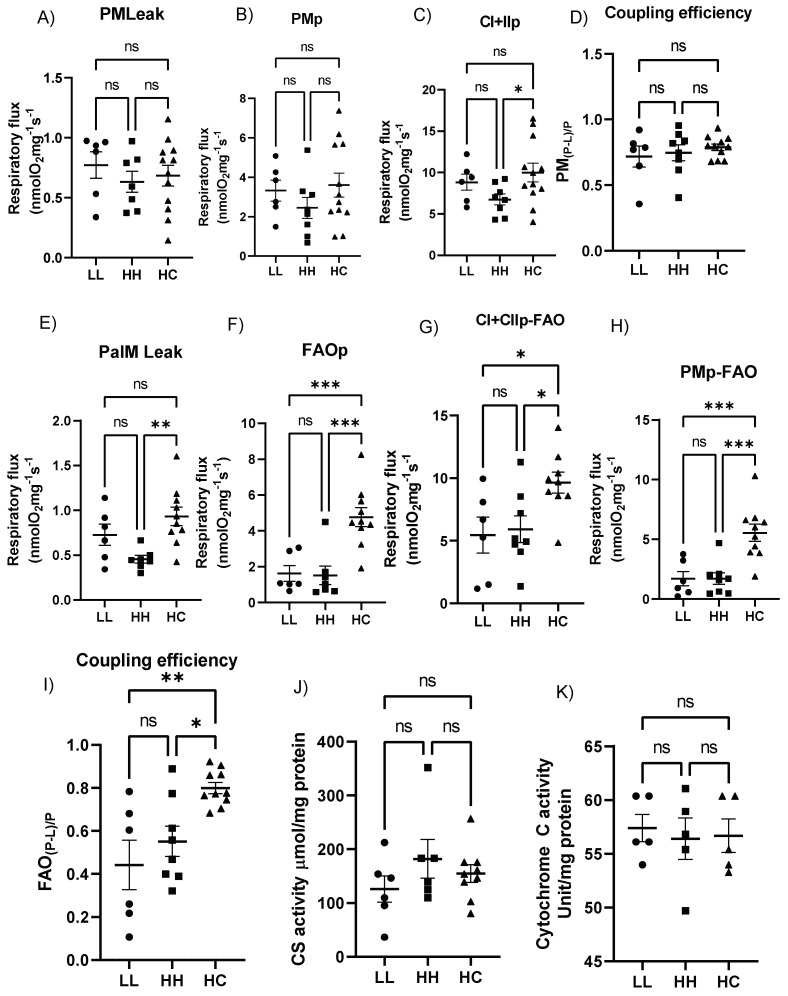
Cardiac mitochondrial respiratory function. (**A**) Electron leak state in the presence of malate and pyruvate; (**B**) Complex I respiration; (**C**) Complex I + II supported OXPHOS; (**D**) Coupling efficiency; (**E**) LEAK respiration in the presence of fatty acid substrates; (**F**) ADP-stimulated respiration rates by pyruvate and palmitoycarnitine; (**G**) Complex I-dependent respiration in addition to malate; (**H**) Fatty acid oxidation with succinate; (**I**) OXPHOS coupling control factors in the presence of fatty acids; (**J**) Citrate synthase activity; and (**K**) Cytochrome C oxidase activity; LL, offspring born from maternal LFLS and postnatal LFLS indicated by closed circles (●); HH, maternal HFHS and postnatal HFHS indicated by closed squares (■); HC, maternal HFHS weaned onto a standard laboratory chow diet indicated by a closed triangle (▲). Values are presented as mean  ±  SEM. Statistical significance is calculated by one-way ANOVA with post hoc Fisher’s LSD comparisons. * *p* < 0.05; ** *p* < 0.01. *** *p* < 0.001; ns, not significant. *n* = 6 in LL; *n* = 7–8 in HH; and *n* = 9–12 in HC.

**Figure 5 metabolites-12-00563-f005:**
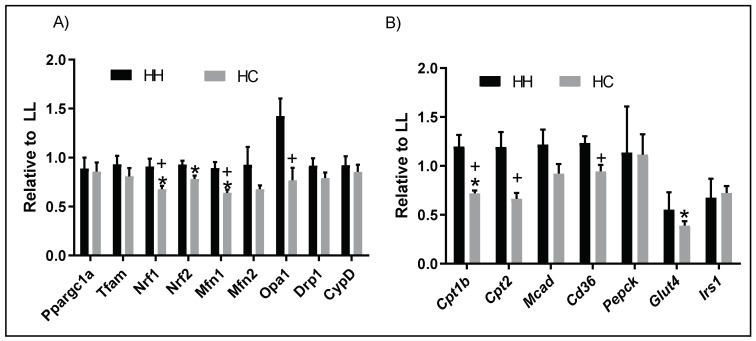
Genes associated with mitochondrial function and metabolism. (**A**) Genes regulating mitochondrial biogenesis and dynamics; and (**B**) Fatty acids and glucose metabolism. *Ppargc1α*, peroxisome proliferator-activated receptor gamma coactivator 1-alpha; *Tfam*, mitochondrial transcription factor A; *Nrf*, nuclear respiratory factor; *Drp1*, dynamic-related protein 1; *Mfn*, dynamic-related GTPase termed mitofusin; *Opa1*, optic atrophy protein 1; *CypD*, peptidylprolyl isomerase D; *Cpt*, creatine palmitoyltransferase; *Cd36*, the surface receptor cluster of differentiation 36; *Glut4*, glucose transporter type 4; *Mcad*, acyl-CoA dehydrogenase medium-chain; *Pepck*, phosphoenolpyruvate carboxykinase, mitochondrial; *Irs1*, insulin receptor substrate 1; LL, offspring born from maternal LFLS and postnatal LFLS; HH, maternal HFHS and postnatal HFHS; HC, maternal HFHS weaned onto a standard laboratory chow diet. Values are presented as mean  ±  SEM as fold change relative to the LL group. Statistical significance is calculated by one-way ANOVA with post hoc Fisher’s LSD comparisons. * *p* < 0.05, vs. LL; + *p* < 0.05, vs. HH. *n* = 4–8 samples/group.

**Figure 6 metabolites-12-00563-f006:**
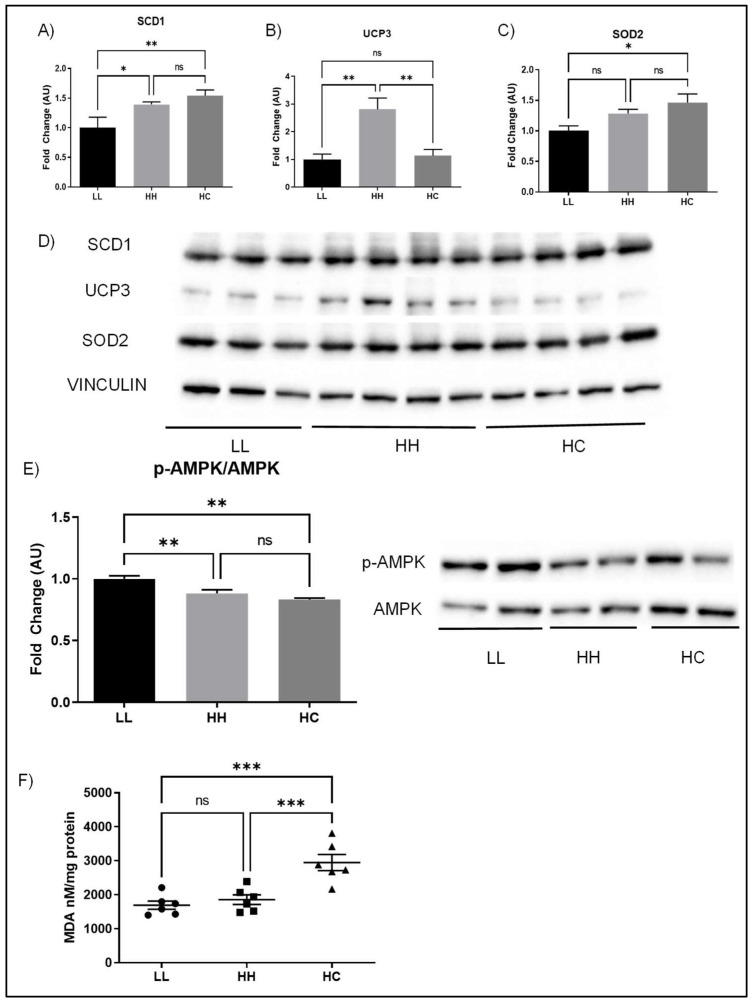
Effects of diet on metabolic proteins. (**A**) SCD1; (**B**) UPC3; (**C**) SOD2; (**D**) the representative blots are shown, and the protein levels were normalized with vinculin; (**E**) p-AMPK/AMPK with a representative blot; and (**F**) MDA, a marker of oxidative stress. SCD1, stearoyl-coenzyme A desaturase 1; UCP3, uncoupling protein 3; SOD3, superoxide dismutase 2, mitochondrial, also known as manganese superoxide dismutase; AMPK, AMP-activated protein kinase; MDA, malondialdehyde; LL, offspring born from maternal LFLS and postnatal LFLS indicated by closed circles (●); HH, maternal HFHS and postnatal HFHS indicated by closed squares (■); HC, maternal HFHS weaned onto a standard laboratory chow diet indicated by a closed triangle (▲). Values are presented as mean  ±  SEM. Statistical significance is calculated by one-way ANOVA with post hoc Fisher’s LSD comparisons. A total of 3–4 animals per group were used with 4 technical replicates per animal for the Western blot and *n* = 6/group was used for measuring MDA levels. * *p* < 0.05; ** *p* < 0.01; *** *p* < 0.001; ns, not significant.

**Figure 7 metabolites-12-00563-f007:**
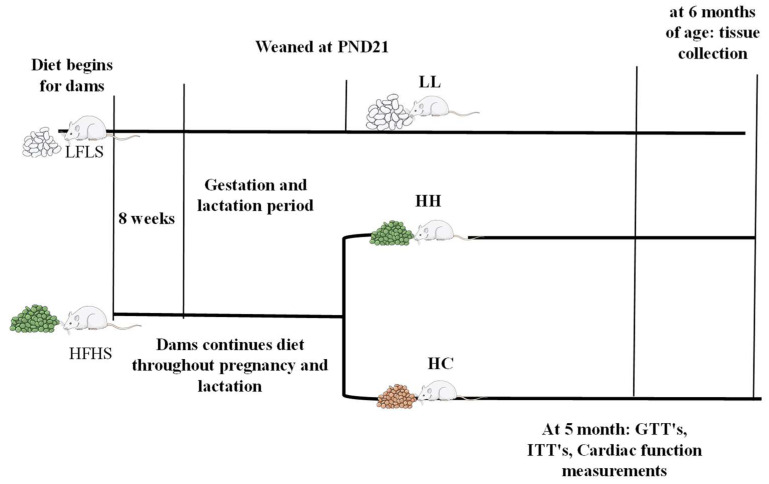
Experimental timeline. Mating was initiated after 8 weeks of diet intervention. LL, offspring born from maternal LFLS and postnatal LFLS; HH, maternal HFHS and postnatal HFHS; HC, maternal HFHS weaned onto a standard laboratory chow diet; PND, postnatal day.

**Table 1 metabolites-12-00563-t001:** Plasma lipid biomarkers and hormones.

	LL	HH	HC
Total cholesterol (mg/dL)	122.3 ± 12.5 (6)	216.1 ± 19.1 * (8)	128.8 ± 7.4 + (8)
LDL (mg/dL)	20.80 ± 0.92 (6)	43.38 ± 5.94 * (8)	22.38 ± 2.02 + (8)
Non-essential fatty acids (µM)	320.8 ± 18.0 (5)	368.1 ± 28.8 (8)	383.7 ± 39.9 (9)
Triglycerides (ng/mL)	149.0 ± 2.8 (5)	108.1 ± 19.6 (8)	113.3 ± 13.8 (8)
Insulin (ng/mL)	4.26 ± 1.08 (6)	17.76 ± 2.51 * (7)	6.53 ± 2.24 + (3)
Leptin (ng/mL)	4.12 ± 1.09 (6)	13.82 ± 1.24 * (7)	12.48 ± 0.52 * (6)

LL, offspring born from maternal LFLS and postnatal LFLS; HH, maternal HFHS and postnatal HFHS; HC, maternal HFHS weaned onto a standard laboratory chow diet. Values are presented as mean ± SEM and sample size is in parenthesis (n). Statistical significance is calculated by one-way ANOVA with post hoc Fisher’s LSD comparisons. * *p*  <  0.05 compared to LL; + *p*  <  0.05 compared to HH.

**Table 2 metabolites-12-00563-t002:** Cardiac function assessed by Echocardiography in 5–6-month-old male offspring.

M-Mode Parameters	LL (*n* = 12)	HH (*n* = 15)	HC (*n* = 12)
LV IVS (cm)			
Diastolic	0.104 ± 0.006	0.105 ± 0.004	0.113 ± 0.005
Systolic	1.154 ± 0.003	0.147 ± 0.006	0.155 ± 0.007
LV PW (cm)			
Diastolic	0.113 ± 0.010	0.119 ± 0.007	0.115 ± 0.007
Systolic	0.164 ± 0.009	0.166 ± 0.010	0.170 ± 0.011
LV diameter (cm)			
Diastolic	0.446 ± 0.011	0.473 ± 0.006 *	0.469 ± 0.012
Systolic	0.304 ± 0.009	0.342 ± 0.012 *	0.332 ± 0.014
RWT	0.47 ± 0.02	0.47 ± 0.02	0.448 ± 0.02
LV, %FS	32.28 ± 1.31	28.83 ± 1.73	30.10 ± 1.86
LV, %EF	64.52 ± 1.88	59.89 ± 3.75	63.81 ± 2.14
HR, bpm	532.8 ± 16.8	523.8 ± 15.9	557.9 ± 13.96

Values are expressed as mean ± SEM. LL, offspring born from maternal LFLS and postnatal LFLS; HH, maternal HFHS and postnatal HFHS; HC, maternal HFHS weaned onto a standard laboratory chow diet; HFHS, a high-fat and a high-sucrose diet; LFLS, a low-fat and a low-sucrose diet; LV, left ventricle; IVS, interventricular septum thickness; PW, posterior wall; RWT, relative wall thickness calculated by the formula [(IVSd + LVPWd)/LVId]; d, diastole; FS, fractional shortening; EF, ejection fraction; HR, heart rate. * *p* < 0.05 compared to LL.

## Data Availability

The data presented in this study are available in article and Appendix A.

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
