# Peer review of "Switching to a Standard Chow Diet at Weaning Improves the Effects of Maternal and Postnatal High-Fat and High-Sucrose Diet on Cardiometabolic Health in Adult Male Mouse Offspring"

_metabolites, 2022, doi:10.3390/metabo12060563_

Round 1
Reviewer 1 Report
good job, my only suggestion would be to be careful with the name of each experimental group because it becomes difficult to understand
Author Response
Response 1: We thank the reviewer for his/her careful reading of our manuscript and appreciate the concern for the experimental group. We have appropriately revised and clarified the experimental group.
Reviewer 2 Report
This study has some issues need to be clarify. The comments are shown in the follows.
1. Lines 1-3. Title. What types of dietary interventions has a modulating effect? “dietary interventions” is a vague term. Does male adult offspring refer to human or mouse offspring? This is also a vague term.
2. Lines 19-22. The hypothesis being tested is unclear. What dietary interventions will be invested? Intervention for whom?
3. Lines 30-31. In the conclusion sentence, what type of diet intervention has an effect on long-term HFFS? What is the meaning about “offspring”?
4. Lines 39-49, 53-58, 373-376. The same question about “offspring” exists in these sentences. The offspring refers to the offspring of humans or mice?
5. Line 60. Please clarify what is “male adult offspring”.
6. Lines 68-71, 227-233. There should not other study results in the Result section, this is the part of the discussion.
7. Lines 72-83. Please explain what are HFHS/OR, HFHS/OP, BW, LFLS, HC, LL, and HH, etc. The logic step for an article display is putting Materials and Methods in front of the results.
8. Lines 102-108, 192-200. In Figures 2 and 5. Some important findings can be presented as numeric evidence, not just graphical representations. For readers, most of the time, we would like to examine and cite the numeric evidence obtained in the research findings.
9. Lines 125-132. In Figure 3. Some important findings were suggested to tabulate their numerical findings in a table.
10. Lines 376-382. Potential mechanisms, the results from the reference [41], and “Improvements in circulating LDL …” are the statements in regard to discussion and the results of other studies, not the conclusion of this study.
11. Line 478, “( CT),” a typo.
12. Line 386, Materials and Methods. Because this study involved maternal high-fat high-sugar diet, offspring metabolic disorders, dietary intervention, etc., can the authors make a timetable diagram of the experimental subjects, analysis content and themes of this experiment to illustrate the entire study picture.
Author Response
Response to Reviewer 2 Comments
This study has some issues need to be clarify. The comments are shown in the follows.
Point 1. Lines 1-3. Title. What types of dietary interventions has a modulating effect? “dietary interventions” is a vague term. Does male adult offspring refer to human or mouse offspring? This is also a vague term.
Response 1: We have changed the title to “Switching to a standard chow diet at weaning improves the effects of maternal and postnatal high-fat and high-sucrose diet on cardiometabolic health in adult male mouse offspring”
Point 2. Lines 19-22. The hypothesis being tested is unclear. What dietary interventions will be invested? Intervention for whom?
Response 2: The change has been made: “We tested the hypothesis that switching to a standard chow diet at weaning would attenuate systemic metabolic disorders, cardiac dysfunction, and mitochondrial dysfunction associated with maternal and postnatal HFHS diet in adult male mouse offspring.”
Point 3. Lines 30-31. In the conclusion sentence, what type of diet intervention has an effect on long-term HFFS? What is the meaning about “offspring”?
Response 3: We used the term “diet intervention” since children born from obese mothers are more likely to eat the same food as their mothers [1], and the postweaning diet was switched to the control diet in the HC group. However, we understand the reviewer’s concern because this term is confusing and can be interpreted as dietary supplementation/ bioactive compounds in addition to a feeding diet. We have deleted “diet intervention” throughout the manuscript and added “mouse” offspring to clarify this study was done using mice, not humans.
- Bayol, S.A.; Farrington, S.J.; Stickland, N.C. A maternal 'junk food' diet in pregnancy and lactation promotes an exacerbated taste for 'junk food' and a greater propensity for obesity in rat offspring. Br J Nutr 2007, 98, 843-851, doi:10.1017/S0007114507812037.
Point 4. Lines 39-49, 53-58, 373-376. The same question about “offspring” exists in these sentences. The offspring refers to the offspring of humans or mice?
Response 4: We added “mouse” and stated adult male mouse offspring throughout the manuscript.
Point 5. Line 60. Please clarify what is “male adult offspring”.
Response 5: We changed to “adult male mouse offspring”.
Point 6. Lines 68-71, 227-233. There should not other study results in the Result section, this is the part of the discussion.
Response 6: We deleted lines 68-71 and 227-233 were moved to the Discussion section.
Point 7. Lines 72-83. Please explain what are HFHS/OR, HFHS/OP, BW, LFLS, HC, LL, and HH, etc. The logic step for an article display is putting Materials and Methods in front of the results.
Response 7: Thank you very much for the suggestion. The abbreviation was given after full explanation.
Point 8. Lines 102-108, 192-200. In Figures 2 and 5. Some important findings can be presented as numeric evidence, not just graphical representations. For readers, most of the time, we would like to examine and cite the numeric evidence obtained in the research findings.
Response 8: GTT AUC and ITT AUC for each group were given numeric values.
Point 9. Lines 125-132. In Figure 3. Some important findings were suggested to tabulate their numerical findings in a table.
Response 9: Figure 3 was replaced with Table 1.
Point 10. Lines 376-382. Potential mechanisms, the results from the reference [41], and “Improvements in circulating LDL …” are the statements in regard to discussion and the results of other studies, not the conclusion of this study.
Response 10. This was moved to the lines 374-376.
Point 11. Line 478, “( ΔΔ CT),” a typo.
Response 11: This change () has been made.
Point 12. Line 386, Materials and Methods. Because this study involved maternal high-fat high-sugar diet, offspring metabolic disorders, dietary intervention, etc., can the authors make a timetable diagram of the experimental subjects, analysis content and themes of this experiment to illustrate the entire study picture.
Response 12: An experimental timeline figure was added to the Material and Methods section.

Reviewer 3 Report
A good and well designed and performed study. The experimental procedures are adequate and the results justify the conclusiones reached. There are only a few language points from which I would advice minor changes for including a revision by an English native colleague.
Author Response
Response 1: We thank the reviewer’s comments. We have corrected all grammatical errors, and have had the manuscript reviewed by three native English speakers.
Round 2
Reviewer 2 Report
None.